# Socio-Economic and Environmental Challenges of Small-Scale Fisheries: Prognosis for Sustainable Fisheries Management in Lake Kariba, Zambia

Imikendu Imbwae [1,2,*], Shankar Aswani [1,3] and Warwick Sauer [1]

1   Department of Ichthyology and Fisheries Science (DIFS), Rhodes University, Makhanda 6140, South Africa
2   Department of Fisheries, Choma P.O. Box 630450, Zambia
3   Department of Anthropology, Rhodes University, Makhanda 6140, South Africa
*   Correspondence: g19i0001@campus.ru.ac.za

**Abstract:** The Lake Kariba fishery is of regional importance; it accounts for 35% of the total Zambian fish production. However, emerging evidence in the recent decades suggests that the fishery is facing socio-economic and environmental challenges. Using Ostrom's framework for analysing socio-ecological systems, we examined the social, economic, and environmental problems faced by the fishing communities in Lake Kariba. The framework links various social, economic, and ecological factors to devise a sustainable fisheries management plan. A combination of survey questionnaires, focus group discussions, observations, and key informant interviews were used to assess this sustainability challenge. The data collected were subjected to bivariate and descriptive analysis. The results obtained did not show a significant decline in fish production over the past 13 years ($R^2$ Linear = 0.119, $p$ = 0.248). However, the experts and the fishers have reported declining trends in valuable fish species such as *Oreochromis mortimeri*, compounded by the increased fishing efforts ($X^2$ = 180.14, $p$ value = < 0.00001). The key threats identified include: overfishing, weak institutions, and the introduction of invasive fish species such as *Oreochromis niloticus*. This situation has raised fears of fish depletion among the stakeholders. Based on these results, we recommend stronger institutional collaboration among the stakeholders in the riparian states and education that illustrates the global value of fisheries for food security and biodiversity conservation in pursuing the United Nations Sustainable Development Goals.

**Keywords:** biodiversity; overfishing; environmental; socio-economic; sustainability

## 1. Introduction

Lake Kariba is a man-made lake composed of two distinct fisheries, namely the small-scale, multi-species inshore fishery and the small pelagic semi-commercial, single-species offshore fishery for the introduced *Limnothrissa miodon* [1]. Lake Kariba accounts for 35% of the total Zambian fish production and 90% of Zimbabwean fish production [2,3]. The contribution of the Lake Kariba fishery to food security has been acknowledged in the recent decades, but its role as a source of employment and income generation to many actors in the country has been grossly underestimated [4]. The contribution of the fishery towards food security is of great importance to the national and regional economy. This narrative makes the fishery a classic 'hot-spot' for fisheries biodiversity and a reliable source of livelihoods for the millions of people subject to the wider anthropogenic threat and conservation priority [1]. The fishery is increasingly under pressure from multiple uses, such as tourism development, recreational fisheries, semi-industrial fishing of the introduced *limnothrissa miodon*, locally known as (kapenta), and, more recently, inland aquaculture through cage fish farming. The multiple use competition has often disadvantaged small-scale fishers by outcompeting them for fishing space [5]. The interactive effect of these human activities can negatively affect the aquatic environment in ways that

lead to low fish productivity, such as by increased sediment loading and increased user conflicts [5–7]. Over the past two decades, there have been wide reports of a declining trend in fish yield and shifts of the fish species composition in Lake Kariba, especially of the more valuable *Oreochromis* species [8,9]. This trend has raised concerns about fish depletion among the local and regional communities [10–12]. However, it is not clear whether this situation has deteriorated or not or how the fisherfolk have adapted to the socio-economic and ecological changes over time [13]. It is assumed that fisheries that are poorly managed will experience a decline in diversity [14]. The management of inland fisheries has not been adequately debated in recent policy deliberations, at either the local or the international levels, which is partly due to the inadequate data to support their value for food and nutritional security [15,16].

In the last 15 years, the Food and Agriculture Organisation (FAO) of the United Nations has facilitated joint international fisheries management meetings among the riparian countries sharing Lake Kariba, meetings which have continued every two years to date [10]. However, the success of these international engagements appears not to have been effective in addressing the conservation aspirations of the local resource users [17,18], which is partly due to weak institutional capacities and management approaches that focus too much attention on fish stock assessment [17]. The lake's socio-economic and environmental data have often remained relatively obscure [17]. This oversight in the acquisition of socio-economic data has resulted in knowledge gaps essential to address food security and poverty among small-scale fishers [19]. Amidst these challenges, climate change appears to exert more pressure by altering the fish habitats and the distribution patterns of these aquatic resources [20,21]. The climatic trends around Lake Kariba show that the temperature will rise in the range of 1–3 °C by 2080, while mean rainfall rates are expected to decline [22]. The unprecedented changes brought about by the COVID-19 pandemic are another stressor constraining conservation efforts [23].

The existing management approaches are under increasing pressure and are unable to cope with and mitigate the rapidly developing pressures such as the environmental impact, including overfishing, the user conflicts, and the invasion of animal and plant fish species [24]. This study, using the Lake Kariba fishery as a case study, contributes to the various studies which emphasise the need to integrate socio-economic data in fisheries management to enhance the sustainability of the inland fisheries [25]. The overall objective of this study was to evaluate the nature of the small-scale fishing practices on Lake Kariba, to examine their socio-economic conditions, and to identify the environmental threats that may have negative effects on the biodiversity of the fisheries and the livelihoods of the fishers. The study also assesses how small-scale fishers have adapted to the socio-economic and environmental changes over time and suggests possible interventions to protect the fishery from further deterioration. The following research questions are addressed:

1. What are the perceived key threats to fisheries productivity in Lake Kariba?
2. How are the socio-economic and environmental conditions in the Lake Kariba fishery affecting small-scale fishers?
3. How have small-scale fishers adapted to the socio-economic and environmental changes in Lake Kariba?
4. How will the management of the Lake Kariba fishery have to change in response to the socio-economic and environmental changes?

## 2. Materials and Methods

### 2.1. Study Site

The study was undertaken in the Zambian section of the Lake Kariba fishery, a social–ecological system that emerged following the construction of a dam wall along the middle of the Zambezi River [24]. The dam was constructed between 1958 and 1961 to generate hydroelectric power (HEP) [24]. The fishery is located along the border between Zimbabwe in the south and Zambia in the north and lies between latitudes 16°28' and 18°06' south and longitudes 26°40' and 29°03' east (Figure 1). By volume, Lake Kariba is the second largest

artificial lake in the world, with an estimated total surface area of 5580 km$^2$ and a length of approximately 280 km, with an altitude of 665 m at the average high-water level [25]. Lake Kariba was made for a single purpose, i.e., hydroelectric power generation for both Zambia and Zimbabwe, but as it turned out, the lake supports a thriving small-scale fishery, hydropower generation, sport fishing, tourism, crocodile farming, and a semi-commercial fishery that contributes significantly to the economies of both Zambia and Zimbabwe and the region at large [5]. The average temperature around Lake Kariba is 28.7 °C. The fishery has vast conducive fishing zones [5,17], favourable for capture fisheries activities. Inland aquaculture production has considerably grown in the recent past, making Zambia the sixth largest producer of farmed fish in Africa [26]. Lake Kariba now houses some of the largest cage fish farms on the African continent, with the production estimate of 17,000 tonnes per year [12]. The two riparian countries, Zambia and Zimbabwe, share the lake at a ratio of 45 to 55, respectively [27]. The Zambian section of Lake Kariba was chosen as a study area because it represents the best-fit example of a fishery that has undergone wide socio-economic and environmental changes over the past five decades with little or no documented papers on the topic of study [25]. A study of this nature will provide the information necessary for improved management and policy direction towards the development of inland fisheries in Zambia and the region at large [13].

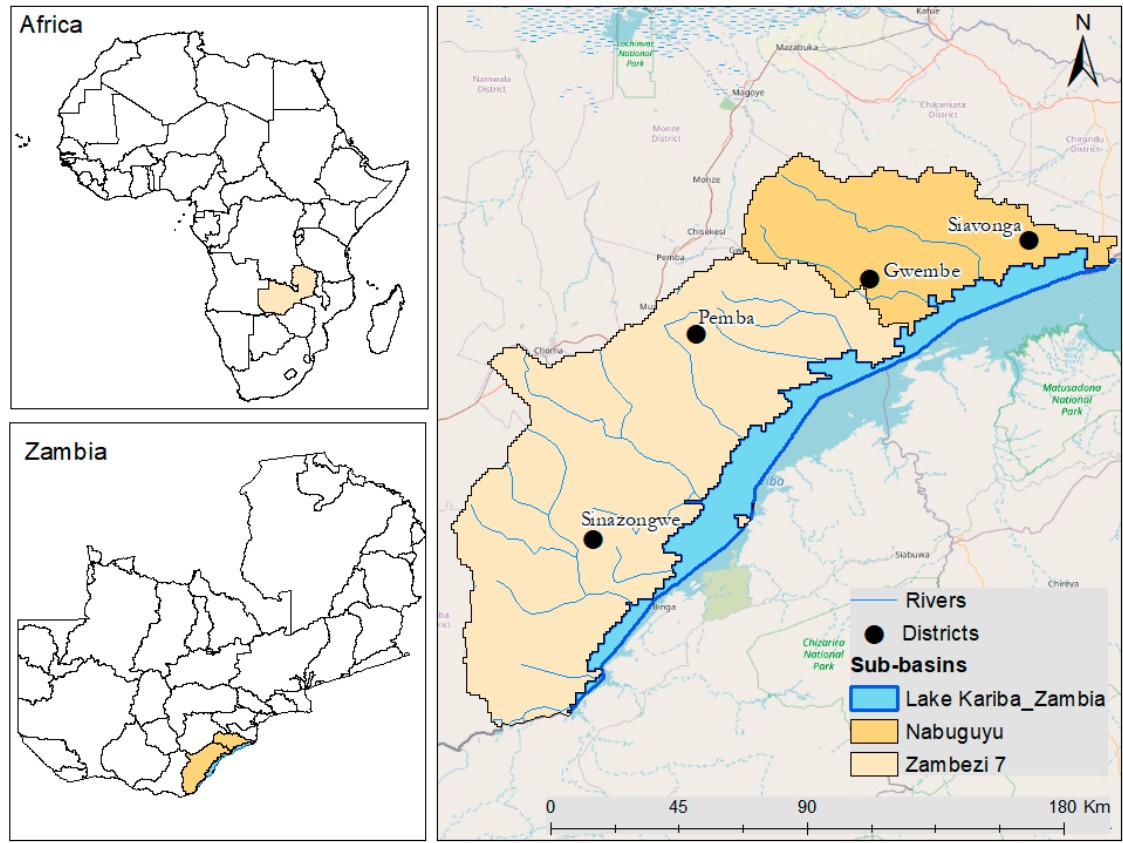

**Figure 1.** Map showing Lake Kariba, Africa and Zambia (Source: own illustration).

In accordance with the principles of Bazigos [28], the lake on the Zambian side is divided into four different strata based on the geographic and ecological features of the lake [17]. These strata fall in three political districts: Sinazongwe, Gweembe, and Siavonga. Each of these districts has a collection of fishing villages ruled by a traditional chief. Ethical clearance to conduct the study was reviewed (2020-1571-4696) and granted by Rhodes university prior to the commencement of the survey. The Department of Fisheries as gate keepers gave us the authority to collect data from their established institutions located in the study area.

## 2.2. Sampling Design

Data collection for this study area (Figure 1) was undertaken from September 2020 to March 2021, using a mixed methods approach to obtain both qualitative and quantitative data [29]. The secondary data (grey literature), such as those from frame survey reports and catch assessment surveys (CAS), were obtained from the Department of Fisheries. This information was vital for the sampling design and a better understanding of the ecological aspect of the fishery with regard to the changes in the fish composition, the fish production trends, and the demographic characteristics of the fishery over time [29]. The Zambian section of Lake Kariba has 65 fishing villages and camps scattered across the shoreline [27].

A reconnaissance survey was conducted prior to the actual data collection to comprehend the status of the fishery and to devise appropriate sampling procedures. This was necessary because the fishing communities are heterogenous, and certain elements of the population sample may not match all the particulars of the previously defined sampling procedures [30]. Thus, to select the sample size in each stratum, a proportional quota sampling technique was used during the study with the help of the secretariat of the fishing village management committee [31]. This technique enables the researcher to obtain sample sizes in each stratum which are directly proportional to the number of participants in a targeted population (Table 1). In order to generalise the results of the study with a minimum level of errors, the study assumed 30% of the total of 65 fishing villages along the entire shoreline [27,32], resulting in a total of 20 fishing villages where the interviews and focus group discussions were carried out (Table 1).

**Table 1.** Sample size population per stratum and district.

| District | Stratum | No. of Fishers Sampled per Stratum | No. of Fishing Villages Sampled | No. of Focus Group Discussions per Stratum |
|---|---|---|---|---|
| Sinazongwe | I | 125 | 6 | **5** |
| | II | 52 | 4 | **3** |
| Gwembe | III | 63 | 5 | **4** |
| Siavonga | IV | 60 | 5 | **4** |
| Total | | 300 | 20 | **16** |

To validate the accuracy of the data collecting tools, 30 questionnaires were initially administered to the local participants in the study area [33,34]. The survey questionnaire focused on fishing practices, environmental and socio-economic conditions such as access to social amenities, livelihood strategies, ecological aspects, and fish marketing strategies. A total of 300 mixed survey questionnaires comprising both open- and closed-ended questions [34] were administered to fishers $\geq$ 20 years old. The selection criteria of the participants (fishers) were put at a 95% confidence level from an estimated total of 2240 fishers across the four strata [35]. When selecting individual participants for the survey questionnaires, a snowball sampling technique was used to identify active fishers in every targeted fishing village [35]. This technique was appropriate for identifying active fishers at the village level since some of the fishers were not available on site during the survey, thus hindering the taking of a random sample [34].

The focus group discussions (FGDs) targeted both fishers and local community members, such as fish processors, members of the public, and fish traders. Each discussion group comprised 9–14 people, thereby covering 162 participants of all genders. Each FGD lasted about three hours and was conducted along the shoreline and in places where the participants could sit properly and be easily observed. The discussions focused on fishing practices, livelihood strategies, general constraints, and the environmental issues affecting the fishery. FGDs are useful for the validation of data and an in-depth understanding of the topic of study [30]. The field observations covered the surrounding environment, and fishing activities across the fisheries value chain, including fishing techniques, waste management forms, and the sewage system, were observed during the field study. Ob-

servations are an effective way of obtaining data regarding the surrounding of the study area [36].

Semi-structured interviews with 18 key informants involving 11 different stakeholders in the fishery were conducted to solicit additional information on the topic of the study. The target group comprised local leaders, experienced fishers, senior government officials, and traditional leaders such as village heads. The key informants helped to confirm and explain the earlier ideas collected from the fishers using questionnaire surveys and FGDs. [33]. A rapid appraisal of fisheries management systems (RAFMS) was used to identify the possible interventions for sustainable fisheries management [31,32]. For the RAFMS, 26 well-informed community members with key informants were identified to help provide information based on their past experience, skills, and knowledge about the fishery [37–39]. The idea was to capture views from a broader perspective to consolidate and validate other sources of information [36,37].

### 2.3. Data Analysis

Before analysis, the data obtained from the survey questionnaires were coded and entered into an Excel spreadsheet for quality control [31]. Descriptive statistics were generated using R Version 3,9.1, 2020. The qualitative data obtained from the interviews and FGDs were analysed using a theme-based content analysis technique, which involved a process of going through all the interviews to categorise themes with similarities [35]. The quantitative data obtained during the survey were analysed using bivariate statistics [32]. As such, the chi-square (Pearson's $\chi2$) test was applied to determine the increase in the fishing effort, using indicators such as the increase in the number of fishers, fishing boats, and nets between 2006 and 2011. This information was the best available information and was vital in analysing the dynamics of the demographic changes in the fishery over time. The fish production trends over the past 13 years were analysed using a curve fitting estimation regression test. This test provides the best fit for specific curves in the dataset for trend analysis over time [32]. All the statistical analyses were performed using the software package R Version 3,9.1 (R Core Team, 2020) to facilitate the interpretation and analysis of the data. The results obtained, together with qualitative data from secondary data sources, were used to describe the socio-economic and environmental condition of the fishery [30,38].

## 3. Results

### 3.1. Fishing Practices

Lake Kariba is a diverse artificial fishery characterised by multiple user groups using a multi-method and multi-species approach to the fishery. According to the study, about 77% of the fishers in Lake Kariba appear to be 'fishing-dependent' (Table 2). However, the dependence degree varies with respect to the type of fisher. The study identified three (3) categories of fishers: full-time, part-time, and seasonal fishers. However, a substantial overlap exists between these fishers. The findings of the study showed that more than half of the fishers (74%) were full-time, 12% were part-time, and 14% were seasonal (Table 2).

**Table 2.** Fisher information based on the 300 survey questionnaires.

| Fishing Information | Part-Time (*n* = 36) | Seasonal (*n* = 40) | Full-Time (*n* = 224) |
|---|---|---|---|
| No. of fishers (% total) | 12% | 14% | 74% |
| Education (% literate) | 22% | 20% | 28% |
| Age (years) | 40 ± 7.49 | 24 ± 6.22 | 56 ± 7.8 |
| Fishing period | 13 ± 3.9 | 15 ± 3.7 | 27 ± 4.9 |
| Gender | - | - | - |
| Male | 24 | 35 | 230 |
| Female | 6 | 5 | - |

The focus group discussion revealed that the average fish catches per unit effort (CPUE) were between 1.5 kg and 2.5 kg for the full-time fishers and 1.5 kg for the part-time fishers. Most of the people in Lake Kariba are either directly or indirectly involved in fishing activities as their main economic stay, full-time activity, or part-time employment. Six types of fishing gear were identified; these included cast nets, drag nets, seine nets, traps, hook and line, and baskets locally known as Miono.

### 3.2. Social Amenities

Regarding social amenities, the study showed that 34.7% of the fishers still use water from the lake for home consumption, while 14.93% indicated having access to and being able to use borehole water. Only 29% confirmed having pit latrines, but 11.28% of the respondents in the study area indicated that they had no access to any form of a toilet. Schooling is problematic too; although more than 65% of the fishing villages have primary schools within a distance of 0 to 5 km, the secondary school distance remains a challenge. The captured data show that more than 70% of the secondary schools can only be accessed at a distance greater than 5 km (FGDs). With regard to health services, 44.7% of the fishing villages fell within a distance of up to 5 km from the nearest clinic; these were mainly from strata IV and II. In stratum I, 40% of the fishing villages were more than 10 km from the nearest clinic.

### 3.3. Economic Status of Fishers

According to the study, the fishers in Lake Kariba dispose of their fish catches on reaching the landing sites. About 74% of the daily fish catches were sold to local consumers and residents, while the remainder (23%) were for consumption. About 3% of the remainder of the fish caught were exchanged for other commodities, such as maize meal, cooking oil, and second-hand clothes. Most of the fishers indicated that the sale of fresh fish to local customers fetched a high price and minimised marketing costs. The average price of fish sold was valued at USD 2.15 per kg in 2021, a price that is over 90% higher than the price of USD 0.960 per kg recorded in the 2011 frame survey. The price of fish depends on the quality, weight, species, and seasonality. The key informants claimed that income from fish produce had fallen owing to seasonal fluctuations and the decline in the valuable *Oreochromis* fish species. The fish catches are perceived to have gradually declined over the last decade, but the increased market values for the fish have not kept pace with this decline. The finding of the study indicated that most of the fishers in the lake (44%) earn about USD 150 per month, while 5% of them earn more than USD 600 a month (Table 3).

**Table 3.** Household monthly income of (fishers).

| Variable | | Frequency (*n* = 300) | Percent % |
|---|---|---|---|
| Monthly household income | <USD 160 | 130 | 43.3% |
| | USD 161–230 | 60 | 20% |
| | USD 231–330 | 45 | 15% |
| | USD 331–430 | 26 | 8.6% |
| | USD 431–530 | 24 | 8% |
| | USD 531–630 | 15 | 5% |
| | USD 630> | - | - |
| Current income compared to 10 years ago | Better | 32 | 10.6% |
| | Stable | 38 | 12.6% |
| | Decreased | 230 | 76.6% |

According to the key informants, the fish prices kept on increasing every year. However, the overall fish production trend shows that the fish production has remained relatively stable. A curve fit estimation regression test carried out revealed that there

had been no significant difference in the fish production trends over the past 13 years ($R^2$ Linear = 0.119, *p* = 0.248) (Figure 2).

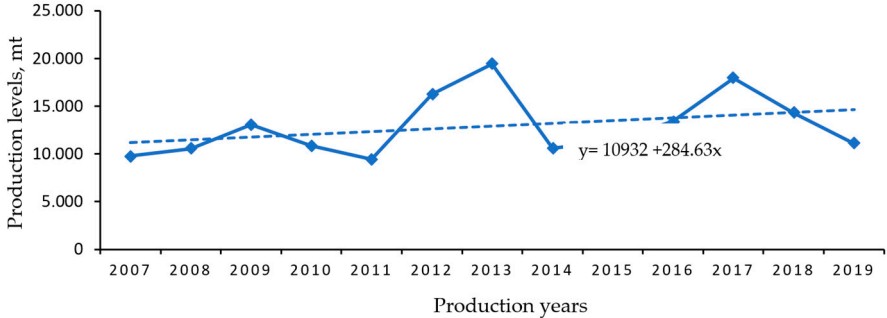

**Figure 2.** Fish production trend (tonnes per annum) in Lake Kariba between 2007 and 2019. Source: DOF [39].

Although the overall fish production in Lake Kariba has remained stable, according to the focus group discussions the small-scale fishers for all categories have not produced the anticipated levels of economic gains from fishing, but fishing has notably helped those referred to here as full-time fishers in sustaining their livelihoods. The part-time fishers have been prevented from sinking into the vicious circle of poverty as their fishing proceeds frequently supplement wages earned from labouring. Small-scale fisheries provide a more reliable source of income for fishers than the wages derived from non-fishing activities.

*3.4. Social Conditions and Fisher Vulnerability*

Our findings show that the fishers are educationally, socially, and economically disadvantaged. They have inadequate access to financial services, and their families face numerous challenges, the most common among them being low income. Irrespective of the different categories, most of the fishers reported that their income from fish sales was spent on purchasing essential commodities, such as clothes, food, and household maintenance. Credit institutions are very limited in the entire fishing community, with only 20% of fishers able to access financial aid in the form of microcredit from government-funded institutions, with interest rates ranging between 25 and 38% per year. Food insecurity in the fishery was observed to be a common problem which the fishers attributed to low income and the high food prices experienced in the recent past. According to the key informants, the prices of food over the past five years had increased by 60%. Most of the fishers interviewed (80%) stated that their household could no longer afford three meals a day and that they had reduced the number of meals, sometimes swapping to undesirable but inexpensive food. The proportions of the fish catch consumed at the household level varied; for example, the full-time fishers consume a much lower proportion (10%) of their fish catch, preferring to sell three-quarters of their catch and buy cheaper foodstuff. Generally, the fishers consume most of their fish catch (85%) and trade the remainder, particularly the large fish, which are perceived to fetch a relatively much higher price. The seasonal fishers tend to consume more than half their fish catch.

The vulnerability context has subjected fishers to low income due to the seasonal price fluctuation of fish catches. Forty percent of the respondents revealed that the fishers were vulnerable to sickness and diseases, especially during the rainy season. The fishers suffer from cholera, dysentery, diarrhoea, malnutrition, and malaria. The study observed that most of the fishing villages rely on water directly from the lake for washing clothes, drinking, and bathing. Other challenges raised by fishers included lack of capital (50%), low income (43%), and user conflicts (44%). The catches were said to be much better in summer than any other season of the year, and most of the respondents indicated that they harvested very low quantities of fish in winter, which gave them a paltry return for most of their daily expenditure (Table 3). Further responses from all the categories of fishers

revealed that user conflict among the various users of the fishery was a widening problem arising from conflicting interest over access to fishing grounds. These user conflicts occur between inland aquaculture farmers, semi-industrial Kapenta fishers, anglers, and small-scale fishers over access to fishing grounds. Conflict often intensifies when small-scale fishers are restricted from the fishing grounds where they have fished for subsistence over decades

### 3.5. Environmental Conditions of the Fishery

According to the study, most of the experienced fishers and experts expressed concern over the decline in the fish species of economic value, such as *Oreochromis mortimeri*. About 70% of the respondents reported that overfishing was among the most important threats to the fisheries. The chi-square test that was performed showed a statistically significant difference in the selected indicators of the fishing efforts between 2006 and 2011. ($X^2$ = 180.14, *p*-value = < 0.00001) (Table 4). Generally, most of the fishery areas in the country have recorded an increase in fishing effort indicators.

**Table 4.** Differences in rate of increase in selected indicators between 2006 and 2011.

| Increase in the Last 5 Years | 2006 | 2011 |
|---|---|---|
| Total number of fishermen | 2804 | 4653 |
| Total number of boats | 2431 | 3451 |
| Total number of nets | 19,500 | 26,769 |

$X^2$ = 180.14, *p*-value = <0.0001. Source: adopted and modified from the LKFRI [27].

According to the study, 71% of the respondents identified land use practice (e.g., subsistence agriculture and human settlements) among the other threats to the fishery (Figure 3). About 76% of the respondents indicated that the operation of the hydroelectric power station can affect fish productivity by altering the distribution and migration pattern. Only 66% of the fishers identified poor legislation as a threat to the fisheries. A further environmental threat identified by the respondents was the deterioration in the water quality due to pollution (Figure 3). Field observation revealed that the waste from human activities such as gardening and subsistence agriculture and excessive loads of nutrients from household waste and municipal sewage are disposed directly into the lake. The main problem is related to the increased nutrient input from household waste and the municipal sewage load directly entering the lake. This was observed to be a common problem, especially in strata IV and II. According to the key informants, organic matter, solid waste, and nutrients that leach into the lake were a result of land use practices and urbanisation. (Figure 3). The introduction of invasive fish species was identified as one of the emerging environmental threats to Lake Kariba (Figure 3).

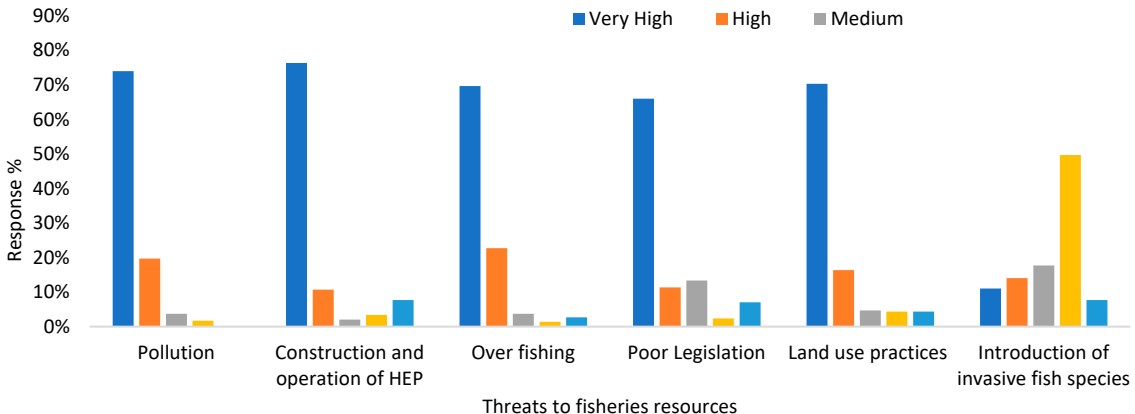

**Figure 3.** Factors perceived by fishers as threats to fisheries resources in Lake Kariba.

### 3.6. Livelihood Strategies

Fishing appears to be the most important source of livelihood across the four strata, with 82% of the respondents undertaking this activity, while 21% are practicing both fishing and fish farming activities. The field observation confirmed these livelihood strategies. Vegetable gardening and small livestock production are some of the livelihood adaptation strategies observed in the study area. The crops grown include millet, sorghum, and maize. Other forms of livelihood strategies observed include activities such as running a shop, selling airtime, and fish processing, which was mostly practiced by women. Generally, most of respondents in Lake Kariba were entirely dependent on fishing. Reducing the number of meals and substituting with less favoured but inexpensive food to save scarce resources were common adaptation strategies. The study further revealed that most fishers have adapted by increasing fishing efforts using multiple fishing nets. The full-time fishers were reported as selling most of their fish catch in order to buy less expensive foodstuffs. According to key informants, some of the fishers from Zambia have opted to cross into the Zimbabwean waters for fishing expeditions, where fish catches are reported to be in abundance.

## 4. Discussion

In the years between 1960 and 1980, the Lake Kariba fishery was robust and rich in fisheries biodiversity [9,40]. In the recent decades, natural and anthropogenic activities have affected the fishery [24,41]. Although the statistical data on fish production from Lake Kariba have not shown a significant decline in the overall fish production over the past 13 years, a decline in both the numbers and the size of the *Oreochromis* fish species has been observed and reported by various authors [8,11,42], as has a general decline in fish species of high economic value *(Oreochromis spp)* [11]. This trend suggests that over-exploitation of a fishery may not only be characterised by a decline in overall production, but also by the shifts in the species composition and the extinction of individual fish species [11]. These changes have raised concerns about fish depletion among the various stakeholders along the agri-fisheries value chain [42,43]. Most of the key informants stated that activities such as overfishing and land use practice have caused considerable ecological concern about the decline in both the water quality and the fisheries. It is generally accepted that the rampant use of destructive fishing gear and the increased fishing efforts are indications of overfishing [43]. The analysis of the frame survey data for Lake Kariba shows that there is a statistically significant increase in the fishing efforts, raising concerns about overfishing. The fishers alleged that small-scale fishing was not destructive to the fishery; they argued that fishing had been going on for decades. However, some of the key informants acknowledged that small-scale fishing has the potential to affect the fishery negatively when destructive fishing gear and methods are used; for instance, the use of draw nets (chasing fish into a gillnet by beating the water) have a negative impact on fish spawning and mouth brooders [14].

The distribution and the composition of key fish species have shifted over time; this is thought to be due to multiple interactive stressors such as overfishing and habitat destruction [40]. The increase in the fishing efforts and the targeting of larger and more valuable fish species can lead to more serious ecological disturbances that can affect the biological diversity of the fisheries in what is known as "fishing down the food web", which involves harvesting relatively larger-sized multi-species fish assemblages and replacing them mainly with small-sized ones at low trophic levels [11,43]. The loss of these valuable fish species has the potential to affect the regeneration processes of the fisheries biodiversity as rich fish species communities are known to promote a more stable environment [11,43]. Even though the fishers on Lake Kariba acknowledge the progressive increase in the fishing efforts, they argue that they have limited livelihood options. This underscores the need to have management strategies that are based on a broad human development perspective through the enhancement of alternative livelihoods, such as aquaculture development and beekeeping, among others [44].

The study identified water pollution as a serious emerging environmental challenge caused by deforestation, mining activities, and subsistence agriculture along the shoreline. These activities have negatively affected the fishery, leading to environmental problems related to water quality, habitat loss, and sedimentation, especially in strata II and IV, where the fishery has experienced rapid urbanisation. Coal mining and agriculture activities along the shoreline have further contributed to the pollution [25]. Organic matter, solid waste, and other forms of nutrients leach into the lake; these substances have the potential to cause significant environmental stress in the fishery and in the oxygen balance of the aquatic environment, with cascading effects on fish productivity and livelihoods [45]. This sustainability challenge can disrupt and alter the composition of fish species, including benthos and planktons, and affect fish productivity [45]. Field observation shows that inland aquaculture is growing fast in Lake Kariba and is generally viewed as an efficient way of scaling up fish productivity and bridging the fish deficit in Zambia, which has remained at 9 kg per capita for over a decade now [26]. Despite this potential, the sector has suffered from a lack of quality support and a clear regulatory framework to guide management perspective [46]. In some areas, it has even led to environmental and social problems, due to a lack of environmental understanding of the linkages between the supporting environment and the cultivation. The commercialisation of the Lake Kariba fishery for inland aquaculture development, especially in stratum IV, has contributed to the loss of productive habitat and reduced the water area for nursery and fish spawning grounds. This development has higher risks in that it may introduce fish diseases such as epizootic ulcerative syndrome (EUS) and the Tilapia Lake virus, among others [47].

The development of specific regulations to govern this industry, especially those related to environmental impact and management, requires further environmental risk assessment and monitoring, which are currently not available for Lake Kariba [46,47]. This development exerts pressure on the limited fishing grounds, with serious implications for the policy reforms in the fisheries and aquaculture sector in Zambia. Our observations on the aspect of the engineering works and operational activities of the hydroelectric power plant on the Lake Kariba fishery revealed negative effects on the hydrological regimes, with fragmentation of the adjacent biodiversity. This scenario has negatively affected fish distribution, with migratory fish species such as *Label altivelis* reportedly being affected. A further environmental challenge reported was the introduction of alien invasive fish species such crayfish (*Cherax quadricarinatus*). The latter is reported to have been accidently introduced into Lake Kariba from fish farm escapees around the 1990s. By 2009, crayfish were widely distributed and had a fully established population throughout the lake. The introduction of the crayfish into Lake Kariba and on the African continent acts as a nuisance, destroying fish eggs, nests, and habitats for spawning fish [48,49]. Most fishermen have complained and reported anecdotally the manner in which crayfish affect their catches. Some of the potential effects include the transfer of diseases and parasites; competition with other crustacea; predation on larvae and fish eggs; and habitat destruction due to functional responses such as burrowing habits and macrophytes cropping. Crayfish can cause considerable damage to fishing gear, resulting in increased servicing costs [50]. The entanglement of crayfish in gillnets has often resulted in ghost fishing gear as the fishers opt to abandon their nets; the partial eating of fish trapped in static gillnets is a source of concern among fishers in Lake Kariba [48]. There has been no serious intervention or biosafety measures to control the spread and invasion of invasive fish species from the relevant authorities in Lake Kariba [46]. Furthermore, the displacement and disappearance of the abundant Kariba bream (*Oreochromis mortimeri*) in Lake Kariba has been attributed to the accidental introduction of the invasive Nile tilapia (*Oreochromis niloticus*) in the 1980s, perpetuating interspecific competition for food and space, agonistic interaction, and intraguild predation [41,51].

While the negative impact of the invasive *Oreochromis niloticus* has widely been documented in the literature [41,48,49], the positive aspect of this fish species as an efficient feed converter has made it acceptable as the most important fish species harvested in Lake

Kariba, comprising 50% of the total catch [41]. *O niloticus* precipitated the development of the aquaculture sector and is now among the most popular sizes of table fish in Zambia [3]. It ranks as the most productive commercially valuable fish species in both capture fisheries and inland aquaculture in Lake Kariba [3,41]. This, together with the successful establishment of *Limnothrissa moidon*, has put Lake Kariba among the best examples of the successful utilisation of introduced fish species for stock enhancement [52]. However, more studies are required to assess the environmental challenges associated with the introduction of *Oreochromis niloticus* and other invasive fish species such as *Limnothrissa miodon.* In addition to the presence of animal invasive species, plant invasive species such as water hyacinth have been established in Lake Kariba, especially in areas around the dam hall [7]. The establishment of this invasive plant species has the potential to condense open areas for fishing, interfere with fishing expeditions, and affect the movement of clogged waterways. The abundance of water hyacinth can reduce the dissolved oxygen content, which may have impact on fisheries biodiversity [52,53].

The present study has further shown that poor legislation has constrained management efforts to enhance fisheries productivity in Lake Kariba. The fisheries policy direction in Zambia currently aims at maximising fish production to meet the urban fish demand and employment opportunities for rural communities [17]. However, the regulations do not limit the amount of fishing gear that an individual fisher can use, and the fishery still operates in an open-access form [17]. This raises concerns as uncapped access can lead to intensive fishing with severe impacts on fish breeding [44,54]. If the current socio-economic and environmental condition in Lake Kariba remains unchecked, it could lead to what Hardin [55], described as the 'Tragedy of the Commons', where unfettered access to common resources, such as littoral fisheries, leads to unchecked exploitation and cascades into deep and wide environmental degradation [55,56].

The Fisheries Act no. 22 of 2011 appears to be comprehensive and provides good regulations for the management of the fisheries. However, it is characterised by the attributes of a centralised management system that undermines the active participation of the local resource users in fisheries management. The Fisheries Act empowers the Fisheries Directorate to initiate and lead the local management plans that are often imposed [57]. Although the Fisheries Act provides for community participation in the management of fisheries resources in Lake Kariba, the level of participation is moderately practiced in the sense that their roles are not well defined, despite the skill, experience, and environmental knowledge fishers bring to participatory management [58]. This has created a knowledge gap in how indigenous knowledge can serve as a foundation on which to extrapolate specific interventions that are contextual in nature [59]. Several studies have shown that active community participation in the management of natural resources is the context in which to overcome environmental challenges and a cost-effective method of resource management [60–62], because the government is often too overwhelmed to reach out to highly scattered areas for monitoring and enforcement of fisheries regulations [63]. The discussions with the key informants revealed that the fishermen on Lake Kariba are often engaged in assisting the government officials to collect statistical data on fish [17]. The scenario underscores the need to have a stronger institutional arrangement that provides for real authority for community participation in the management of their resources to legitimise stakeholder participation [64,65].

In light of the socio-economic and environmental challenges faced by the small-scale fishers on Lake Kariba, adaptation strategies exist among the fishers; some fishing households seem to diversify their livelihoods towards subsistence agriculture and livestock production, while others remain as fishers; some increase fishing efforts, while others are engaged in casual duties such as the selling of basic commodities in marketplaces. However, these adaptation strategies do not provide a stable source of income [57]. Agriculture may be a stronger alternative livelihood, but the topography and climatic conditions of the area appear to favour very few crops, such as drought resistance crops [5]. The rainfall season is short and low (less than 500 m) due to climatic changes [66]. These climatic changes are

likely to have social, economic, and environmental impacts on fisheries and agriculture, with cascading implications for livelihoods, food security, and income generation. Ndebele-Murisa et al. [22] predicted that temperatures around the Lake Kariba catchment area have been rising faster than the IPCC forecasts for Southern Africa. These environmental problems have been exacerbated by the impact of land tenure transformations which occurred in the 1950s, before and after the dam wall construction [24]. The construction of the dam wall in the middle of the Zambezi River, forming what is today known as Lake Kariba, resulted in the mass displacement of over 57, 000 indigenous Tonga/Korekore ethnic people [24]. Their resettlement was forced and badly planned [67,68], and those displaced communities are still suffering from the challenges associated with this resettlement that took place in the late 1950s [25]. The state displaced them from the banks of the fertile Zambezi River, where they used to have two seasons of crop production and access to fisheries throughout the year. During this period, three-quarters of the local communities were forcedly relocated to a relative higher and drier arid land, in some cases where there was competition for space with the wildlife animals in adjacent game parks [5]. The consequence of the forced relocation was the loss of a more stable livelihood opportunity. A detailed study on this aspect may be required to draw lessons on the implications of land tenure transformation for livelihoods and freshwater ecosystems. Based on the results of this study, it is apparent that the current environmental pressures may deepen if not adequately addressed; this may have long-term effects on fisheries biodiversity, with adverse impacts on food security [67]. The problem exposes fishers to nutritional food insecurity since most of the people living along the shoreline are highly dependent on the lake for sustenance [68].

## 5. Prognosis for Sustainable Fisheries Management

Despite the existence of two distinct fisheries in Lake Kariba, the living conditions of the fishers is vulnerable to fish decline and the fragile condition of the lake. The living conditions and the socio-economic status of the small-scale fishers has remained poor and subject to pervasive poverty. A major challenge is that the context of the vulnerability of the fishers is not within their control because of government interventions that are often not inclusive of local participation [69]. Therefore, it is very important to devise remedial measures aimed at reducing the level of vulnerability and enhancing resilience in the management of the fishery [61,70]. To enhance the livelihoods of the small-scale fishers and to reduce the impact of fishing pressure on the lake, the management strategies must be able to address the various socio-economic and environmental challenges affecting the fishing communities and the lake upon which the source of their livelihoods highly depends [71]. The idea of sustainable fisheries management provides for various linkages across the multiple social, economic, and ecological facets of the fishery systems [72]. Mutual linkages among these systems may create a sense of responsibility for community participation and may create institutional responsibility [66]. A social-ecological approach which emphasises the human–environment interaction is needed. Ahmed et al. [44] proposed a management system that links various social, economic, and ecological aspects to balance societal needs and objectives. The system highlights a range of ecosystem benefits and services that communities derive from the environment and reconciles the feedback loop relationship of human economics and the ecosystem [44].

We recommend such a management system for Lake Kariba (Figure 4). The economic aspects in this case include fish marketing, fishing costs, income, and employment. The social aspects refer to the livelihoods of the fishers, the cultural factors, and the social benefits. The ecological consideration includes the condition of the lake over the long term, including siltation, pollution, and the conservation of fisheries biodiversity. A comprehensive understanding of how these aspects interact in time and space are essential and must be given consideration [73]. The connections existing between the ecological, social, and economic aspects are shown in Figure 4, and analysing them in this fashion clearly illustrates the influence and importance of the ecological–economic, socio-ecological,

and socio-economic links [61]. The knowledge about this interaction is critical for the conservation of the ecosystem processes that support the fisheries and for the refinement of the social mechanisms for the governance and equal distribution of the economic benefits at different scales [61]. A compressive social and ecological system can withstand the negative shocks and subsequently lessen the vulnerability of the fishers to environmental challenges. [62,74]. This approach constitutes a concept which emphasises the human–environment interactions in the management of fisheries [75]. Fishing is an ancient occupation and a source of food security whose foundation is rooted in human and natural linkages from which economic benefits such as income and employment are derived [62]. The socio-ecological resilience of a fishery is to a large extent dependent on the livelihood security of the users [62]. Ostrom [70] identified four characteristics of a social and ecological system: a resource system (e.g., fishery), a resource unit (fish), the users (fishers), and the governance systems and other interested stakeholders. The interaction of these systems at different levels can be used to produce positive results [70]. In developing an efficient fisheries management structure, it is highly crucial to secure a balance between the environmental, social, and economic benefits and costs to meet the human needs for both present and future generations [68]. Such interventions guarantee sustainable natural resource utilisation such as fisheries and enhance the livelihoods of dependent communities [66]. Three major components that can be used to design a sustainable fisheries management plan have been identified in this study: (i) institutional collaboration, (ii) laws and policies, and (iii) active community participation (Table 5).

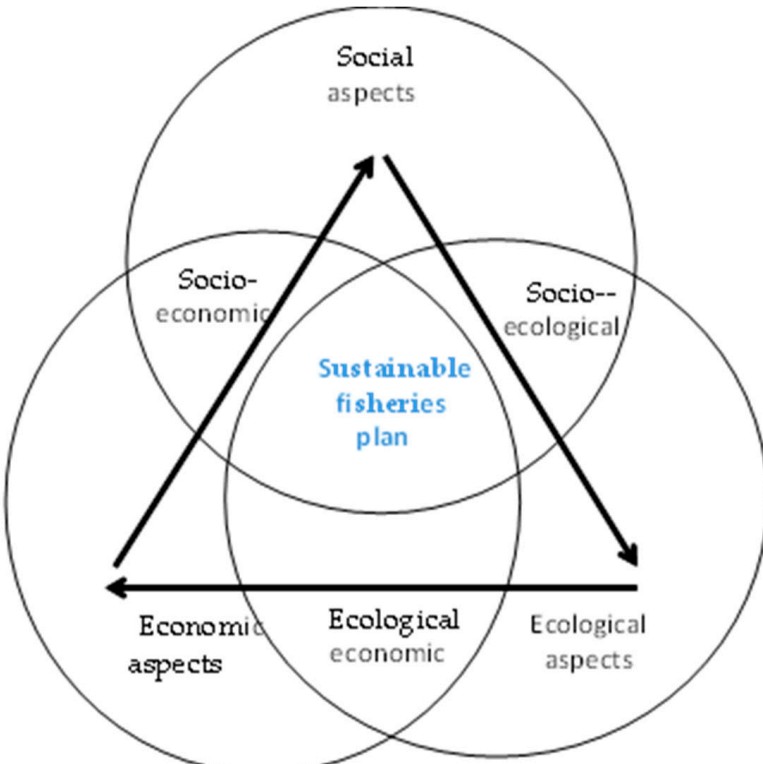

**Figure 4.** A systematic diagram for sustainable fisheries management, illustrating the need to evaluate and account for socio-ecological, ecological–economic and socio-economic interaction in fisheries management. Source: adopted and modified from (Ahmed et al. [44]) and FAO [76].

**Table 5.** Key strategies for sustainable fisheries management.

| Components | Examples |
| --- | --- |
| Institutions | ○ Support systems by both government and private institutions such as NGOs and civil society organisations<br>○ Training services for fishers to enhance capacity and willingness to actively participate in fisheries management<br>○ Institute regular research and monitoring activities for lake management |
| Laws and policies | ○ Pertinent government laws and policies and implementation of fisheries regulations (regulate overfishing and prohibit bad fishing practices)<br>○ Conservation of fisheries biodiversity by way of establishing protected areas such as sanctuaries<br>○ Strengthen environmental protection activities to control pollution and siltation |
| Community participation | ○ Community participation through establishment of fishing village committees<br>○ Active fisher community participation for fisheries resource utilisation and management<br>○ Community awareness programmes for environmental protection and maintenance of lake ecosystems |

Source: adopted and modified from (Ahmed et al. [44]).

The aforementioned aspects will strengthen the capacities of both the local and the national institutions and create partnerships across the institutions of the riparian countries, with a focus on scaling up alternative livelihoods. Furthermore, educational efforts can show the local and international value of fisheries for food security and biodiversity conservation, especially of the commercially valuable fish species [77,78]. Challenges of the freshwater fisheries in this study are attributed to the environmental degradation, which is due to human activities [79,80]. We suggest that public institutions, civil society organisations, and other interested stakeholders jointly working together would aid the protection of the lake environment. The sharing of experience and the exchange of ideas, training, and technical support would generate important knowledge, which is essential for management [81]. The symptoms of the decline in fish resources in Lake Kariba are attributed to multiple factors, but weak policy legislation is, however, very apparent. It is a situation that has been largely characterised by a moderately co-management system of governance [63]. Kapembwa et al. [67] observed that the set-up was so because responsibility for the fisheries management was carried out by government authorities through local authorities instead of the fishing community, thereby defeating the original expectation which was to decentralise the fisheries management. Therefore, the enactment of policies aimed at providing guidelines for resource users' roles under co-management would enhance the conservation measures for fisheries biodiversity [72]. These would further help conserve pure brood stock to scale up aquaculture production, which has currently put Zambia in the lead as the largest producer of farmed fish in Southern Africa [26] and may have positive benefits for the fisher communities [82,83].

Active community participation has been identified as being among the best strategies for achieving sustainable resource management [69] Being a homogenous entity, community participation is considered to be a viable institution for effective collaborative resource management [52]. The assumption is that if community participation in conservation is effective, the benefits accrued from conservation will create a sense of responsibility and ownership for the community members so they can become good stewards of the resources [84–86]. We therefore suggest that a stronger community-based fisheries management structure could be a significant innovation in addressing the aforementioned socio-economic and environmental challenges facing the small-scale fishers in Lake Kariba. This will require a policy direction to provide their roles and responsibilities to ensure

efficiency. It will furthermore help to accelerate the attainment of the United Nations Sustainable Development Goal number 17, which recognises the role of partnerships for sustainable development [84,87]. It is grounded on the hypothesis that strong partnerships will activate a broad range of stakeholders in the acquisition of knowledge, experience, technology, and resources towards attaining the global agenda. The positive outcome of such a strong partnership is the sustainable conservation of biodiversity to alleviate poverty through the responsible fisheries management of the inland fisheries [76].

## 6. Conclusions

The focus of this study was to understand the socio-economic conditions of small-scale fishers and to identify the environmental threats that may have negative effects on fisheries biodiversity and the livelihoods of the fishers in Lake Kariba, Zambia. The results obtained showed that the communities around Lake Kariba depend on fisheries as a source of income and employment. While the overall fish production has not significantly reduced in the past 13 years, fish species of high economic value such as (*Oreochromis* spp.) have declined. The fishery is exposed to widescale environmental threats, such as pollution and overfishing. This trend has negatively affected the livelihoods of the small-scale fishers. The prognosis for sustainable fisheries management in Lake Kariba will depend on the possibility to reconcile the various social, economic, and ecological aspects of the fishery. Such interventions must ensure that the proceeds of enhanced management are shared equitably among the resource users to maintain the long-term stability of their resources. The fishers in Lake Kariba are vulnerable to resource degradation cascading into the loss of social, economic, and ecological proceeds that can be attained through responsible fisheries. To devise a sustainable fisheries management plan, the interacting social, economic, and ecological facets must be considered. Consequently, any effective management intervention must aim at regulating the utilisation of the resources to ensure sustainable use. Hence, a combination of socio-economic, ecological–economic, and socio-ecological approaches to fisheries governance must be incorporated into the management plans. Furthermore, effective enforcement of legal legislation and stronger institutional collaboration among the stakeholders in the riparian states, as well as active community participation in the governance of the resources, will be vital in securing responsible fisheries on Lake Kariba. Furthermore, actual studies on the ecological experiments and other socio-cultural issues are required for further policy recommendation on the topic of study.

**Author Contributions:** I.I. was involved in the field data collection, analysis, and conceptualisation of the study. S.A. gave advice on the scoping, reviewed and edited the paper. W.S. guided the data analysis and the design of the paper. Resources were provided by Zambia Aquaculture Development Project (ZAEDP) and Rhodes University All authors have read and agreed to the published version of the manuscript.

**Funding:** The study was funded partly by the Zambia Aquaculture Enterprise Development Project (ZAEDP) grant number 2000200000602 and Rhodes University.

**Institutional Review Board Statement:** The study was reviewed and approved by the Rhodes University Human Ethics Committee (RU-HEC). Reference number 2020-1571-4696.

**Informed Consent Statement:** Consent was obtained from all the subjects involved in the study.

**Data Availability Statement:** The data presented in this study can be obtained on request from the corresponding author; the data have not been publicised due to ethical restrictions.

**Acknowledgments:** The authors wish to thank the Department of Fisheries in Zambia for providing secondary data and a conducive environment for data collection. We further acknowledge the hard work of Aurthurtone Jere, Bornwel Seeman, Maulu Sahya and Oliver J Hasimuna for proofreading the paper Many thanks go to Harris Phiri for the words of encouragement.

**Conflicts of Interest:** There is no conflicting interest reported by the authors.

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
