# Peer review of "Socio-Economic and Environmental Challenges of Small-Scale Fisheries: Prognosis for Sustainable Fisheries Management in Lake Kariba, Zambia"

_sustainability, doi:10.3390/su15043179_

Round 1

Reviewer 1 Report

This paper covers a very relevant topic ‘’ Socio-economic and environmental challenges of small-scale fisheries: prognosis for sustainable fisheries management in Lake Kariba, Zambia ‘’. This contribution merits publication, but it suffers of too many weaknesses to be accepted as such.

The following is my comments:

1.            Write your manuscript in Journal template.

2.            Continues line number must be added in manuscript.

3.            Abstract is not properly written. Authors are advise to add some sentence about methodology in Abstract.

4.            Objective does not write in proper way in the end of introduction, Write your objectives in points.

5.            The text of this paper in general needs a thorough review, as there are multiple spelling and grammatical errors. Many sentences do not mean any sense. Moreover, there are several sloppy errors that should be fixed.

6.            Method section needs more clarification about why the researchers selected the

7.             in Lake Kariba, Zambia ?

8.            Results and discussion section are poorly written. Discussion section needs more attention.

9.            Discussion part is abounding in citing research that are not very relevant to the area of interest. Some of them are general in nature which need to be pruned.

10.        Discussion is too long and there are some unnecessary contents in the discussion of this article that can be deleted. It is suggested to modify them carefully and refine the main contents of the article again.

11.        Write main results and future recommendation in conclusion. The authors should rewrite the conclusion with more clarity and suggest some mitigation measures to combat the problem.

12.        Reference does not meet to journal style, set these.

Overall, the study conducted is interesting but a major revision of the entire manuscript is essentially required for publication in this journal. Hence, I recommend reconsideration after a major revision of the manuscript.  

Author Response

Response to Reviewer 1 Comments

Comments and Suggestions for Authors (Reviewer 1).

This paper covers a very relevant topic ‘’ Socio-economic and environmental challenges of small-scale fisheries: prognosis for sustainable fisheries management in Lake Kariba, Zambia ‘’. This contribution merits publication, but it suffers of too many weaknesses to be accepted as such.

The following is my comments:

  1. Write your manuscript in Journal template.

Point 1: we have used the standard format and style for MDPI in terms of the font type and size. However, we are unable to realign the title and affiliation appropriately, this may require the help from the editors.

  1. Continues line number must be added in manuscript.

Point 2: We have addressed this important observation by inserting the line numbers

  1. Abstract is not properly written. Authors are advise to add some sentence about methodology in Abstract.

Point 3: we have addressed this important point, by adding a statement in the abstract on how data was analyzed and treated, see line number 16 in the main text.

  1. Objective does not write in proper way in the end of introduction, Write your objectives in points.

Point 4: The objectives have been streamlined and clearly outlined in a statement format. We have outlined the overall objective following a common approach that most authors from this journal (MDPI) uses.

  1. The text of this paper in general needs a thorough review, as there are multiple spelling and grammatical errors. Many sentences do not mean any sense. Moreover, there are several sloppy errors that should be fixed.

Point 5:  We have noticed these errors and addressed the spelling, grammatical errors and other sloppy errors have been corrected in the entire text. See the tracked changes in red

  1. Method section needs more clarification about why the researchers selected Lake Kariba, Zambia?

Point 7: This section of the manuscript has been revised to indicate why Lake Kariba was chosen as a study area. The reasons being advanced for choosing Lake Kariba are that the study area specifically the Zambian section represents the best fit example of a fishery that has undergone a wide socio-economic and environmental changes over the past five decades, (1960-10) after the construction of the dam wall across the Zambezi River. Furthermore, the study area has  no documented study on the topic of study. A study of this nature will provide information for improved management and guide policy direction for the development of inland fisheries in Zambia and the region at large. See line number 111 to 115

  1. Results and discussion section are poorly written. Discussion section needs more attention.

Point 8: We have addressed this comment by removing some of the statements in the results which appeared like discussions and pushed them to discussion section. We have further streamlined the discussion by supporting the hanging statements with references.

  1. Discussion part is abounding in citing research that are not very relevant to the area of interest. Some of them are general in nature which need to be pruned.

We have corrected the mistakes in this section by consolidating it with relevant peer reviewed papers, and streamed certain statements. See red tracked red in the discussion and other reference

  1. Discussion is too long and there are some unnecessary contents in the discussion of this article that can be deleted. It is suggested to modify them carefully and refine the main contents of the article again.

Point 10: The discussion may appear to be long because this kind of study is a first one of its kind in this area or in the region, most studies are focused on fish stock assessment, hydrobiology and biology of individual fish species, post and present socio-economic and environmental issues that emerged after the construction of the dam have not been addressed in literature.

  1. Write main results and future recommendation in conclusion. The authors should rewrite the conclusion with more clarity and suggest some mitigation measures to combat the problem.

Point 11 Major findings and future recommendations have been highlighted in the conclusion, refer to line number 604-610

  1. Reference does not meet to journal style, set these.

Point 12 Mistakes in the reference section have been corrected and highlighted in tract changes

 Overall, the study conducted is interesting but a major revision of the entire manuscript is essentially required for publication in this journal. Hence, I recommend reconsideration after a major revision of the manuscript.  

 Thank you very much for reviewing the paper, comments and suggestions.

Response to Reviewer 1 Comments

Comments and Suggestions for Authors (Reviewer 1).

This paper covers a very relevant topic ‘’ Socio-economic and environmental challenges of small-scale fisheries: prognosis for sustainable fisheries management in Lake Kariba, Zambia ‘’. This contribution merits publication, but it suffers of too many weaknesses to be accepted as such.

The following is my comments:

  1. Write your manuscript in Journal template.

Point 1: we have used the standard format and style for MDPI in terms of the font type and size. However, we are unable to realign the title and affiliation appropriately, this may require the help from the editors.

  1. Continues line number must be added in manuscript.

Point 2: We have addressed this important observation by inserting the line numbers

  1. Abstract is not properly written. Authors are advise to add some sentence about methodology in Abstract.

Point 3: we have addressed this important point, by adding a statement in the abstract on how data was analyzed and treated, see line number 16 in the main text.

  1. Objective does not write in proper way in the end of introduction, Write your objectives in points.

Point 4: The objectives have been streamlined and clearly outlined in a statement format. We have outlined the overall objective following a common approach that most authors from this journal (MDPI) uses.

  1. The text of this paper in general needs a thorough review, as there are multiple spelling and grammatical errors. Many sentences do not mean any sense. Moreover, there are several sloppy errors that should be fixed.

Point 5:  We have noticed these errors and addressed the spelling, grammatical errors and other sloppy errors have been corrected in the entire text. See the tracked changes in red

  1. Method section needs more clarification about why the researchers selected the
  2. in Lake Kariba, Zambia?

Point 7: This section of the manuscript has been revised to indicate why Lake Kariba was chosen as a study area. The reasons being advanced for choosing Lake Kariba are that the study area specifically the Zambian section represents the best fit example of a fishery that has undergone a wide socio-economic and environmental changes over the past five decades, (1960-10) after the construction of the dam wall across the Zambezi River. Furthermore, the study area has  no documented study on the topic of study. A study of this nature will provide information for improved management and guide policy direction for the development of inland fisheries in Zambia and the region at large. See line number 111 to 115

  1. Results and discussion section are poorly written. Discussion section needs more attention.

Point 8: We have addressed this comment by removing some of the statements in the results which appeared like discussions and pushed them to discussion section. We have further streamlined the discussion by supporting the hanging statements with references.

  1. Discussion part is abounding in citing research that are not very relevant to the area of interest. Some of them are general in nature which need to be pruned.

We have corrected the mistakes in this section by consolidating it with relevant peer reviewed papers, and streamed certain statements. See red tracked red in the discussion and other reference

  1. Discussion is too long and there are some unnecessary contents in the discussion of this article that can be deleted. It is suggested to modify them carefully and refine the main contents of the article again.

Point 10: The discussion may appear to be long because this kind of study is a first one of its kind in this area or in the region, most studies are focused on fish stock assessment, hydrobiology and biology of individual fish species, post and present socio-economic and environmental issues that emerged after the construction of the dam have not been addressed in literature.

  1. Write main results and future recommendation in conclusion. The authors should rewrite the conclusion with more clarity and suggest some mitigation measures to combat the problem.

Point 11 Major findings and future recommendations have been highlighted in the conclusion, refer to line number 604-610

  1. Reference does not meet to journal style, set these.

Point 12 Mistakes in the reference section have been corrected and highlighted in tract changes

 Overall, the study conducted is interesting but a major revision of the entire manuscript is essentially required for publication in this journal. Hence, I recommend reconsideration after a major revision of the manuscript.  

 Thank you very much for reviewing the paper, comments and suggestions.

Response to Reviewer 1 Comments

Comments and Suggestions for Authors (Reviewer 1).

This paper covers a very relevant topic ‘’ Socio-economic and environmental challenges of small-scale fisheries: prognosis for sustainable fisheries management in Lake Kariba, Zambia ‘’. This contribution merits publication, but it suffers of too many weaknesses to be accepted as such.

The following is my comments:

  1. Write your manuscript in Journal template.

Point 1: we have used the standard format and style for MDPI in terms of the font type and size. However, we are unable to realign the title and affiliation appropriately, this may require the help from the editors.

  1. Continues line number must be added in manuscript.

Point 2: We have addressed this important observation by inserting the line numbers

  1. Abstract is not properly written. Authors are advise to add some sentence about methodology in Abstract.

Point 3: we have addressed this important point, by adding a statement in the abstract on how data was analyzed and treated, see line number 16 in the main text.

  1. Objective does not write in proper way in the end of introduction, Write your objectives in points.

Point 4: The objectives have been streamlined and clearly outlined in a statement format. We have outlined the overall objective following a common approach that most authors from this journal (MDPI) uses.

  1. The text of this paper in general needs a thorough review, as there are multiple spelling and grammatical errors. Many sentences do not mean any sense. Moreover, there are several sloppy errors that should be fixed.

Point 5:  We have noticed these errors and addressed the spelling, grammatical errors and other sloppy errors have been corrected in the entire text. See the tracked changes in red

  1. Method section needs more clarification about why the researchers selected the
  2. in Lake Kariba, Zambia?

Point 7: This section of the manuscript has been revised to indicate why Lake Kariba was chosen as a study area. The reasons being advanced for choosing Lake Kariba are that the study area specifically the Zambian section represents the best fit example of a fishery that has undergone a wide socio-economic and environmental changes over the past five decades, (1960-10) after the construction of the dam wall across the Zambezi River. Furthermore, the study area has  no documented study on the topic of study. A study of this nature will provide information for improved management and guide policy direction for the development of inland fisheries in Zambia and the region at large. See line number 111 to 115

  1. Results and discussion section are poorly written. Discussion section needs more attention.

Point 8: We have addressed this comment by removing some of the statements in the results which appeared like discussions and pushed them to discussion section. We have further streamlined the discussion by supporting the hanging statements with references.

  1. Discussion part is abounding in citing research that are not very relevant to the area of interest. Some of them are general in nature which need to be pruned.

We have corrected the mistakes in this section by consolidating it with relevant peer reviewed papers, and streamed certain statements. See red tracked red in the discussion and other reference

  1. Discussion is too long and there are some unnecessary contents in the discussion of this article that can be deleted. It is suggested to modify them carefully and refine the main contents of the article again.

Point 10: The discussion may appear to be long because this kind of study is a first one of its kind in this area or in the region, most studies are focused on fish stock assessment, hydrobiology and biology of individual fish species, post and present socio-economic and environmental issues that emerged after the construction of the dam have not been addressed in literature.

  1. Write main results and future recommendation in conclusion. The authors should rewrite the conclusion with more clarity and suggest some mitigation measures to combat the problem.

Point 11 Major findings and future recommendations have been highlighted in the conclusion, refer to line number 604-610

  1. Reference does not meet to journal style, set these.

Point 12 Mistakes in the reference section have been corrected and highlighted in tract changes

 Overall, the study conducted is interesting but a major revision of the entire manuscript is essentially required for publication in this journal. Hence, I recommend reconsideration after a major revision of the manuscript.  

 Thank you very much for reviewing the paper, comments and suggestions.

Response to Reviewer 1 Comments

Comments and Suggestions for Authors (Reviewer 1).

This paper covers a very relevant topic ‘’ Socio-economic and environmental challenges of small-scale fisheries: prognosis for sustainable fisheries management in Lake Kariba, Zambia ‘’. This contribution merits publication, but it suffers of too many weaknesses to be accepted as such.

The following is my comments:

  1. Write your manuscript in Journal template.

Point 1: we have used the standard format and style for MDPI in terms of the font type and size. However, we are unable to realign the title and affiliation appropriately, this may require the help from the editors.

  1. Continues line number must be added in manuscript.

Point 2: We have addressed this important observation by inserting the line numbers

  1. Abstract is not properly written. Authors are advise to add some sentence about methodology in Abstract.

Point 3: we have addressed this important point, by adding a statement in the abstract on how data was analyzed and treated, see line number 16 in the main text.

  1. Objective does not write in proper way in the end of introduction, Write your objectives in points.

Point 4: The objectives have been streamlined and clearly outlined in a statement format. We have outlined the overall objective following a common approach that most authors from this journal (MDPI) uses.

  1. The text of this paper in general needs a thorough review, as there are multiple spelling and grammatical errors. Many sentences do not mean any sense. Moreover, there are several sloppy errors that should be fixed.

Point 5:  We have noticed these errors and addressed the spelling, grammatical errors and other sloppy errors have been corrected in the entire text. See the tracked changes in red

  1. Method section needs more clarification about why the researchers selected the
  2. in Lake Kariba, Zambia?

Point 7: This section of the manuscript has been revised to indicate why Lake Kariba was chosen as a study area. The reasons being advanced for choosing Lake Kariba are that the study area specifically the Zambian section represents the best fit example of a fishery that has undergone a wide socio-economic and environmental changes over the past five decades, (1960-10) after the construction of the dam wall across the Zambezi River. Furthermore, the study area has  no documented study on the topic of study. A study of this nature will provide information for improved management and guide policy direction for the development of inland fisheries in Zambia and the region at large. See line number 111 to 115

  1. Results and discussion section are poorly written. Discussion section needs more attention.

Point 8: We have addressed this comment by removing some of the statements in the results which appeared like discussions and pushed them to discussion section. We have further streamlined the discussion by supporting the hanging statements with references.

  1. Discussion part is abounding in citing research that are not very relevant to the area of interest. Some of them are general in nature which need to be pruned.

We have corrected the mistakes in this section by consolidating it with relevant peer reviewed papers, and streamed certain statements. See red tracked red in the discussion and other reference

  1. Discussion is too long and there are some unnecessary contents in the discussion of this article that can be deleted. It is suggested to modify them carefully and refine the main contents of the article again.

Point 10: The discussion may appear to be long because this kind of study is a first one of its kind in this area or in the region, most studies are focused on fish stock assessment, hydrobiology and biology of individual fish species, post and present socio-economic and environmental issues that emerged after the construction of the dam have not been addressed in literature.

  1. Write main results and future recommendation in conclusion. The authors should rewrite the conclusion with more clarity and suggest some mitigation measures to combat the problem.

Point 11 Major findings and future recommendations have been highlighted in the conclusion, refer to line number 604-610

  1. Reference does not meet to journal style, set these.

Point 12 Mistakes in the reference section have been corrected and highlighted in tract changes

 Overall, the study conducted is interesting but a major revision of the entire manuscript is essentially required for publication in this journal. Hence, I recommend reconsideration after a major revision of the manuscript.  

 Thank you very much for reviewing the paper, comments and suggestions.

Response to Reviewer 1 Comments

Comments and Suggestions for Authors (Reviewer 1).

This paper covers a very relevant topic ‘’ Socio-economic and environmental challenges of small-scale fisheries: prognosis for sustainable fisheries management in Lake Kariba, Zambia ‘’. This contribution merits publication, but it suffers of too many weaknesses to be accepted as such.

The following is my comments:

  1. Write your manuscript in Journal template.

Point 1: we have used the standard format and style for MDPI in terms of the font type and size. However, we are unable to realign the title and affiliation appropriately, this may require the help from the editors.

  1. Continues line number must be added in manuscript.

Point 2: We have addressed this important observation by inserting the line numbers

  1. Abstract is not properly written. Authors are advise to add some sentence about methodology in Abstract.

Point 3: we have addressed this important point, by adding a statement in the abstract on how data was analyzed and treated, see line number 16 in the main text.

  1. Objective does not write in proper way in the end of introduction, Write your objectives in points.

Point 4: The objectives have been streamlined and clearly outlined in a statement format. We have outlined the overall objective following a common approach that most authors from this journal (MDPI) uses.

  1. The text of this paper in general needs a thorough review, as there are multiple spelling and grammatical errors. Many sentences do not mean any sense. Moreover, there are several sloppy errors that should be fixed.

Point 5:  We have noticed these errors and addressed the spelling, grammatical errors and other sloppy errors have been corrected in the entire text. See the tracked changes in red

  1. Method section needs more clarification about why the researchers selected the
  2. in Lake Kariba, Zambia?

Point 7: This section of the manuscript has been revised to indicate why Lake Kariba was chosen as a study area. The reasons being advanced for choosing Lake Kariba are that the study area specifically the Zambian section represents the best fit example of a fishery that has undergone a wide socio-economic and environmental changes over the past five decades, (1960-10) after the construction of the dam wall across the Zambezi River. Furthermore, the study area has  no documented study on the topic of study. A study of this nature will provide information for improved management and guide policy direction for the development of inland fisheries in Zambia and the region at large. See line number 111 to 115

  1. Results and discussion section are poorly written. Discussion section needs more attention.

Point 8: We have addressed this comment by removing some of the statements in the results which appeared like discussions and pushed them to discussion section. We have further streamlined the discussion by supporting the hanging statements with references.

  1. Discussion part is abounding in citing research that are not very relevant to the area of interest. Some of them are general in nature which need to be pruned.

We have corrected the mistakes in this section by consolidating it with relevant peer reviewed papers, and streamed certain statements. See red tracked red in the discussion and other reference

  1. Discussion is too long and there are some unnecessary contents in the discussion of this article that can be deleted. It is suggested to modify them carefully and refine the main contents of the article again.

Point 10: The discussion may appear to be long because this kind of study is a first one of its kind in this area or in the region, most studies are focused on fish stock assessment, hydrobiology and biology of individual fish species, post and present socio-economic and environmental issues that emerged after the construction of the dam have not been addressed in literature.

  1. Write main results and future recommendation in conclusion. The authors should rewrite the conclusion with more clarity and suggest some mitigation measures to combat the problem.

Point 11 Major findings and future recommendations have been highlighted in the conclusion, refer to line number 604-610

  1. Reference does not meet to journal style, set these.

Point 12 Mistakes in the reference section have been corrected and highlighted in tract changes

 Overall, the study conducted is interesting but a major revision of the entire manuscript is essentially required for publication in this journal. Hence, I recommend reconsideration after a major revision of the manuscript.  

 Thank you very much for reviewing the paper, comments and suggestions.

Reviewer 2 Report

However, some concerns are as follows,

1. The clear hypothesis and research questions are absent. 

2. The methodology section is huge and can be reduced. 

3. The result section contains irrelevant discussions which can be transferred to the discussion section.

4. Some figures and concepts presented here might have copyright issues. Rather than present in the manuscript, the model can be well discussed in texts. 

5. The conclusion is huge and can be modified into 4-5 sentences. 

6. Reference section is majorly based on reports, and can be modified into research papers. 

7. The references are not aligned with the journal style. 

The rest of the comments can be found in the comment section of the manuscript file.

Thank you very much. 

Author Response

Response to reviewer 2

 Comments and Suggestions for Authors

However, some concerns are as follows,

  1. The clear hypothesis and research questions are absent. 

Point 1: This comment has been addressed; we have since outlined the research questions in the manuscript.

  1. The methodology section is huge and can be reduced. 

Point 2:  The methodology appears to be huge because data were collected using multiple data-collection tools and techniques. (Tringulation) For example, semi-structured questionnaires, focus group discussions (FGDs), Rapid appraisal fisheries management system (RAFMS), key informants’ interviews and observations. Multiple data-collection tools were used for triangulation purposes.

Triangulation refers to using multiple methods or data sources in research to develop a comprehensive understanding of phenomena and data validation. Some of the ecological data is too old, it was therefore necessary to validate such information using multiple data sources. According to FAO code of conduct for responsible fisheries, we need to use the best available information to generate knowledge as a best management practice [87]. However, we have attempted to streamline and justify the methodology used in this study.

  1. The result section contains irrelevant discussions which can be transferred to the discussion section

We have addressed this comment, certain statements in the results which appeared like discussions have been moved to discussion section see line number 219 to 224 has been moved to discussion section

  1. Some figures and concepts presented here might have copyright issues. Rather than present in the manuscript, the model can be well discussed in texts.

Point 4: To avoid copyright issues, the model has been adopted and modified in the manuscript. Furthermore, we have highly acknowledged the authors of the model under caption and intext reference. See line number 530-539

  1. The conclusion is huge and can be modified into 4-5 sentences.

The conclusion has been streamlined by presenting the major findings, future study area fucus, of the study and mitigation to remediate the situation

  1. Reference section is majorly based on reports, and can be modified into research papers.

We have further consolidated the reference section with peer reviewed papers. Out of the 87 references, listed 65 are pear reviewed papers and 22 are technical reports. 

  1. The references are not aligned with the journal style. 

 We have made corrections to realign the reference with the journal style of reference.

The rest of the comments can be found in the comment section of the manuscript file.

We have taken note of the comments in the manuscript and corrected the mistakes and errors for example, certain data appears to be a bit outdated. We applied the principle of FAO code of conduct for responsible fisheries which requires us to use the best available data as the basis for fisheries management. This is a fundamental requirement for precautionary approach and good management in general.

Round 2

Reviewer 1 Report

All comments are completed. Now it is ready for publish.